# A Theoretical Solution for Analyzing Bi-Layer Structures with Differing Thermal Properties [note 1]

**DOI:** 10.3390/mi16121320

**Published:** 2025-11-25

**Authors:** Qianhua Peng, Siyuan Zhou, Yan Shi, Xiaohui Qian

**Affiliations:** 1State Key Laboratory of Mechanics and Control for Aerospace Structures, Nanjing University of Aeronautics & Astronautics, Nanjing 210016, China; sz2301185@nuaa.edu.cn (Q.P.); yshi@nuaa.edu.cn (Y.S.); 2H3C Technology Co., Ltd., No.8 Guangshunnan St., Beijing 100102, China; zhou.siyuana@h3c.com

**Keywords:** solar cell, thermal load, theoretical solution, simplified model

## Abstract

Based on the Hilbert–Riemann theory, this paper develops a simplified model to address interfacial fracture in bi-layer laminated solar cells with significantly dissimilar thermal properties. The model is used to analyze interfacial normal stress distributions and identify critical stress points, taking into account the substantial mismatch in the coefficients of thermal expansion between the semiconductor and encapsulation layers. The predicted temperature and stress fields are validated through finite element simulations. Furthermore, by investigating commonly used encapsulation films and solar cell modules, the coupled effects of the thermal expansion coefficient and elastic modulus are elucidated. The results demonstrate that, under a constant layer thickness, the position of the stress critical point is governed by two dimensionless parameters: the ratio of thermal expansion coefficients and the ratio of elastic moduli. This work offers an efficient and practical approach for predicting thermal stress concentration trends in laminated solar cell structures, thereby providing useful insights for the design and fabrication of solar modules.

## 1. Introduction

Adhesive bonding is widely used in microelectronic device assembly and has proven to be a robust and reliable method for integrating semiconductors with encapsulating layers, particularly under mild operating conditions. However, its performance degrades significantly in extreme environments such as space. A notable example is found in space solar cell manufacturing, where this technique bonds materials with substantially mismatched thermomechanical properties. For instance, the commonly used encapsulant ethylene-vinyl acetate copolymer (EVA) exhibits a coefficient of thermal expansion about 35 times greater than that of silicon, while its Young’s modulus is roughly four orders of magnitude lower. When subjected to the extreme temperature variations characteristic of the space environment (approximately −190 °C to 160 °C), this severe property mismatch induces significant stress concentrations at the adhesive interfaces.

This phenomenon has drawn considerable interest from both scientists and engineers. A study by Marlene Knausz et al. [1] on nine encapsulating materials revealed that three exhibited significant shrinkage prior to expansion during heating. As solar modules endure diurnal and seasonal thermal cycles, the differential thermal expansion among their components generates internal stress [2,3], which can lead to various failures such as ruptured cells, delamination of the encapsulant, and cracks at interfaces. Selecting an encapsulation material with the appropriate thermo-mechanical properties is thus an effective strategy for suppressing these failure modes [2].

To uncover this fracture mechanism, the distribution of thermal stresses within solar panels composed of different materials is our primary concern. Kislyuk developed an analytical method based on tensile testing and validated its correlation with material aging prediction [4]. Amina Ennemri [5] systematically examines the impact of microcracks on the electrical performance of silicon solar cells and summarizes the most commonly used techniques for crack detection. Zafar Ullah Khan [6] critically assesses the performance reliability and degradation of solar PV systems, highlighting the pivotal role of data-driven deep learning and machine learning techniques in analyzing failure mechanisms and advancing predictive maintenance for more accurate financial and technical planning. Khangamlung Kamei [7] demonstrates that thermo-mechanical loads significantly affect fatigue crack propagation and modal parameters, with heating rate and temperature distribution playing critical roles while revealing fundamental differences between pre-seeded and natural crack behaviors. Felix Haase [8] study reveals that the orientation of cell interconnect ribbons relative to the bending axis critically influences crack initiation in solar modules, with parallel ribbons reducing bending tolerance by four times compared to perpendicular ones, while the presence of a backsheet reduces crack width by 30%.

A significant body of research has been devoted to interfacial crack problems in composite materials since the 1960s, aimed at understanding their fundamental mechanics [9]. Herrmann et al. [10] employs the finite element method and a crack propagation criterion based on energy release rate to systematically analyze the nucleation and propagation behavior of thermally induced cracks in dual-phase composite materials. Several researchers, including Tsamasphyros and Theocaris [11], Chen [12], Suo [13], and Linkov [14], have investigated crack problems in perfectly bonded bi-materials under mechanical loading. In the work of Petrova and Herrmann [15], based on the expression of complex potential, a solution to the internal crack system and interface crack problem is constructed. However, the dissimilar properties is seldom considered in the previous analytical works and the derivations are unfriendly to engineers.

To analyze thermal stress in laminated solar cells under thermal loads, this paper proposes a simplified analytical model capable of predicting stress distributions, with validation based on the Hilbert–Riemann theoretical solution for interfacial cracks. This study is also an extension of the previous article of our group [16]. Some discussions of thermal stress and interfacial failure in solar laminates are presented. The study is structured as follows: Following a brief introduction in Section 1, a simplified model of a representative bi-layer structure with highly dissimilar thermal properties is developed in Section 2. This analytical model is then validated in Section 3 through finite element analysis (FEA). Furthermore, the influence of various material parameters is examined in Section 4. Finally, conclusions and remarks are provided in Section 5.

## 2. Theoretical Model

### 2.1. Heat Transfer Analyses

The theoretical model, schematically shown in Figure 1, is formulated for an adiabatic crack lying along the interface of two semi-infinite half-planes. The temperature distribution in this bi-material system is described by the functions *T*_*j*_(*x*, *y*), where the index *j* = 1, 2 identifies the upper and lower half-planes, respectively. The boundary conditions for the temperature perturbation *T*_*j*_(*x*, *y*) are specified as follows:(1)$$k_{1}\frac{\partial T_{1}(x,0^{+})}{\partial y}=k_{2}\frac{\partial T_{2}(x,0^{-})}{\partial y},\left|x\right|\geq a_{0},y=0$$(2)$$k_{1}\frac{\partial T_{1}(x,0^{+})}{\partial y}=k_{2}\frac{\partial T_{2}(x,0^{-})}{\partial y}=0,\left|x\right|<a_{0},y=0$$(3)$$T_{1}(x,0^{+})=T_{2}(x,0^{-}),\left|x\right|\geq a_{0},y=0$$(4)$$\begin{array}{l} k_{1}\frac{\partial T_{1}(x,y)}{\partial y}=k_{2}\frac{\partial T_{2}(x,y)}{\partial y}=q,x^{2}+y^{2}\to \infty \\ k_{1}\frac{\partial T_{1}(x,y)}{\partial x}=k_{2}\frac{\partial T_{2}(x,y)}{\partial x}=0,x^{2}+y^{2}\to \infty \end{array}$$

Let *k* represent the thermal conductivity. The subscript “1” and “2” denote the upper and lower planes, respectively. The function *T_j_*(*x*, *y*) is given by the real part of the complex potential function *f_j_*(*z*).(5)Tj(x,y)=Re[fj(z)]

Introducing the derivative *F_j_*(*z*) as
(6)$$F_{j}(z)=f_{j}^{\prime }(z)$$

And the expressions of the partial derivatives of the function *T_j_*(*x*, *y*) with respect to the variables *x* and *y* are as follows:(7)$$\frac{\partial T_{j}(x,y)}{\partial x}=\frac{1}{2}[F_{j}(z)+\bar{F_{j}(z)}],\;\frac{\partial T_{j}(x,y)}{\partial y}=-\frac{1}{2}[F_{j}(z)-\bar{F_{j}(z)}]$$

For the boundary conditions at the material interface (Equation (1)) to be satisfied, the heat flux must be continuous, thereby ensuring(8)$$k_{1}F_{1}(x)+k_{2}\bar{F_{2}}(x)=k_{2}F_{2}(x)+k_{1}\bar{F_{1}}(x)$$

To account for the thermally insulated condition at the crack faces, a standard analytic continuation is applied as follows:
(9)$$\begin{array}{c} k_{1}F_{1}(z)=-k_{2}\bar{F_{2}}(z),z\in (1)\\ k_{2}F_{2}(z)=-k_{1}\bar{F_{1}}(z),z\in (2) \end{array}$$

By combining the aforementioned method with the temperature continuity condition at the interface (Equation (3)), a function *Q*(*z*) is defined, which is analytic in the entire complex plane save for the crack lines:(10)$$Q(z)=\left\{\begin{array}{c} F_{1}(z)-\bar{F_{2}}(z),z\in (1)\\ F_{2}(z)-\bar{F_{1}}(z),z\in (2) \end{array}\right.$$

Enforcing the adiabatic crack boundary conditions on the combined solution of Equations (9) and (10) reduces the problem to the following Hilbert problem:(11)$$Q^{+}(x)+Q^{-}(x)=0,\left|x\right|<a_{0}$$

A finite crack is situated on the x-axis over the interval (−*a*_0_, *a*_0_). Following Muskhelishvili’s approach [17], the homogeneous solution to Equation (11) takes the form:(12)$$\chi (z)=\sqrt{\left(z-a_{0}\right)}\cdot \sqrt{\left(z+a_{0}\right)}$$

The branch cuts are defined along the crack lines to ensure that the product function for each finite crack decays as 1/*z* at infinity. Consequently, the solution *Q*(*z*) to Equation (11) is given by
(13)$$Q(z)=\frac{C_{0}+C_{1}z}{\sqrt{z^{2}-a_{0}^{2}}}$$
where the coefficients *C*_1_ and *C*_0_ can be obtained from the infinity and single-value conditions, respectively.(14)$$C_{1}=-\frac{iq(k_{1}+k_{2})}{k_{1}k_{2}},C_{0}=0$$

From the solution for *F_j_*(*z*) in Equations (9) and (10), we have(15)$$F_{j}(z)=-\frac{iq}{k_{j}}\cdot \frac{z}{\sqrt{z^{2}-a_{0}^{2}}}\;\;\;\;\;\;\begin{array}{c} \left(j=1,2\right) \end{array}$$

Additionally, the temperature distribution *T_j_*(*x*, *y*) can be directly obtained from Equations (5) and (6).

### 2.2. Thermal Stress Analyses

Turning to the thermoelastic problem shown in Figure 1, the corresponding stress and displacement fields can be formulated in terms of complex functions as follows [18]:(16)$$\sigma_{jy}-i\tau_{jxy}=\Phi_{j}(z)+\bar{\Phi_{j}(z)}+z\bar{\Phi_{j}^{\prime }(z)}+\bar{\Psi_{j}(z)}$$(17)$$2\mu_{j}(u_{j}+v_{j})=\kappa_{j}\phi_{j}(z)-z\bar{\Phi_{j}(z)}-\bar{\psi_{j}(z)}+\beta_{tj}\int f_{j}(z)dz$$(18)$$\phi_{j}^{\prime }(z)=\Phi_{j}(z),\psi_{j}^{\prime }(z)=\Psi_{j}(z)(j=1,2)$$
where$$\kappa_{j}=\left\{\begin{array}{c} 3-4\upsilon_{j}\;\;\begin{array}{c} for\;plain\;strain \end{array}\\ \frac{3-4\upsilon_{j}}{1+\upsilon_{j}}\;\;\begin{array}{c} for\;plain\;stress \end{array} \end{array}\right.\;\;and\;\;\beta_{tj}=\left\{\begin{array}{c} (1+\upsilon_{j})\alpha_{tj}E_{tj}\;\;\begin{array}{c} for\;plain\;strain \end{array}\\ \alpha_{tj}E_{tj}/(1+\upsilon_{j})\;\;\begin{array}{c} for\;plain\;stress \end{array} \end{array}\right.$$

Here, *μ_j_* and *E_j_* represent the shear modulus and Young’s modulus, respectively, while *υ_j_* and *α_tj_* denote Poisson’s ratio and the thermal expansion coefficient.

Then, the function Ω*_j_*(*z*) comes from(19)$$\Omega_{j}(z)=-\Phi_{j}(z)-z\Phi_{j}^{\prime }(z)-\Psi_{j}(z)$$

Substituting the function Ω*_j_*(*z*) into Equations (16) and (17) yields the following:(20)$$\sigma_{j}-i\tau_{j}=\Phi_{j}(z)-\bar{\Omega_{j}(z)}+(z-\bar{z})\bar{\Phi_{j}^{\prime }(z)}$$(21)$$2\mu_{j}\frac{\partial }{\partial x}(u_{j}+iv_{j})=K_{j}\Phi_{j}(z)+\bar{\Omega_{j}(z)}-(z-\bar{z})\bar{\Phi_{j}^{\prime }(z)}+\beta_{tj}f_{j}(z)$$

Considering the free boundaries in Figure 1, we have(22)$${\left(\sigma_{1}-i\tau_{1}\right)}^{+}={\left(\sigma_{2}-i\tau_{2}\right)}^{-},\left|x\right|>a_{0},y=0$$(23)$${\left(u_{1}+iv_{1}\right)}^{+}={\left(u_{2}+iv_{2}\right)}^{-},\left|x\right|>a_{0},y=0$$(24)$${\left(\sigma_{1}-i\tau_{1}\right)}^{+}={\left(\sigma_{2}-i\tau_{2}\right)}^{-}=0,\left|x\right|\leq a_{0},y=0$$(25)$$\sigma_{ij}\to 0,x^{2}+y^{2}\to \infty$$

The stress boundary condition (Equation (22)) stipulates continuity of stress across the interface, which requires(26)$$\Phi_{1}(x)+\bar{\Omega_{2}}(x)=\Phi_{2}(x)+\bar{\Omega_{1}}(x)$$

For convenience of representation, we introduce a new function *B*(*z*) to represent Equation (26)(27)$$B(z)=\left\{\begin{array}{c} \Phi_{1}(z)+\bar{\Omega_{2}}(z),z\in (1)\\ \Phi_{2}(z)+\bar{\Omega_{1}}(z),z\in (2) \end{array}\right.$$

Since the function is holomorphic on the entire plane, it follows from Liouville’s theorem [19] and the remote boundary condition (Equation (25)) that(28)$$\begin{array}{l} \Phi_{1}(z)=-\bar{\Omega_{2}}(z),z\in (1)\\ \Phi_{2}(z)=-\bar{\Omega_{1}}(z),z\in (2) \end{array}$$

Given the continuity of both stress and displacement across the interface, we can accordingly introduce a holomorphic function *D*(*z*) following the same approach.(29)$$D(z)=\left\{\begin{array}{c} \frac{K_{1}}{\mu_{1}}\Phi_{1}(z)-\frac{1}{\mu_{2}}\bar{\Omega_{2}}(z)+\frac{\beta_{t1}}{\mu_{1}}f_{1}(z),z\in (1)\\ \frac{K_{2}}{\mu_{2}}\Phi_{2}(z)-\frac{1}{\mu_{1}}\bar{\Omega_{1}}(z)+\frac{\beta_{t2}}{\mu_{2}}f_{2}(z),z\in (2) \end{array}\right.$$

The combination of Equations (27)–(29) leads to the following Hilbert problem:(30)$$D^{+}(x)+\omega D^{-}(x)=\frac{\beta_{t1}}{\mu_{1}}f_{1}(x)+\omega \frac{\beta_{t2}}{\mu_{2}}f_{2}(x)$$(31)$$\omega =\frac{\Gamma +K_{1}}{\Gamma K_{2}+1},\Gamma =\frac{\mu_{1}}{\mu_{2}}$$

A finite crack is situated along the x-axis within the interval (−*a*_0_, *a*_0_). Following Muskhelishvili’s approach [16], the homogeneous solution to Equation (30) can be expressed as(32)$$X(z)={\left(z+a_{0}\right)}^{\frac{1}{2}-i\varepsilon }{\left(z-a_{o}\right)}^{\frac{1}{2}+i\varepsilon }$$(33)$$\theta =\frac{\Gamma (K_{2}-1)-(K_{1}-1)}{\Gamma (K_{2}+1)+(K_{1}+1)},\varepsilon =\frac{1}{2\pi }\ln(\frac{1-\theta }{1+\theta })=\frac{\ln\omega }{2\pi }$$

With the branch cuts selected along the crack lines to impose the asymptotic condition that the product behaves as 1/*z* for large *z*, the solution to Equation (30) becomes(34)$$D(z)=\frac{1}{2\pi iX(z)}\int_{-a_{0}}^{a_{0}}\frac{X^{+}(t)H(t)dt}{t-z}+\frac{P(z)}{X(z)}$$(35)$$H(t)=\frac{\beta_{t1}}{\mu_{1}}f_{1}(t)+\omega \frac{\beta_{t2}}{\mu_{2}}f_{2}(t)$$

The form of the polynomial *P*(*z*) depends critically on the remote thermo-mechanical coupling conditions. Consequently, as this complex coupling is not known a priori, it is standard practice to set *P*(*z*) = 0.

## 3. Simulation Analyses

A representative FE model was established in ABAQUS using two-dimensional four-node linear quadrilateral elements. The model represents a bi-layer structure of uniform dimensions, featuring a 300 μm interfacial crack—substantially smaller than the 3 mm plate length—with a thickness-to-crack ratio exceeding 10. The steady-state heat transfer and ensuing thermal stresses were computed using dedicated analysis procedures. Furthermore, mesh sensitivity was rigorously assessed to confirm the result accuracy.

The validity of the finite element model presented in Figure 2a was verified against the theoretical solution using a PDMS/silicon bi-layer structure as an example. In this model, the upper layer is PDMS with a thermal expansion coefficient of 2.15 × 10^−4^/°C and a thermal conductivity of 0.21 W/(m·°C) while the lower layer is silicon with corresponding values of 6 × 10^−6^/°C and 148 W/(m·°C), respectively. The thickness of both layers is 150 μm. A uniform heat flux of 0.05 W/m^2^ was applied to the top surface. An axisymmetric model was employed for simplicity, with the coordinate origin set at the crack center and the crack tips located at *x* = ±*a*_0_. The resulting temperature and normal stress fields are shown in Figure 2b and Figure 2c, respectively. A direct comparison between the theoretical and FEA results is provided in Figure 3, demonstrating good agreement.

Leveraging the theoretical results of Section 2, we analyze the temperature fields for different dimensionless ratios of crack width to length (*H*/*a*_0_) and distance from the crack tip as *x*/*a*_0_, respectively. Figure 3a reveals a monotonic convergence of the upper-crack-surface temperature profiles toward the theoretical curve with an increasing value of *H*/*a*_0_, indicating that the thermal field evolves toward the semi-infinite plane solution. It can be foreseen that when the thickness–length ratio is big enough, the temperature distribution along the upper surface of the crack will be completely consistent with the theoretical curve. For *H*/*a*_0_ bigger than 0.8, the simulation results closely match the predictions of the theoretical model. Furthermore, the model’s validity is supported by its consistency with realistic operational conditions.

The interfacial stress field is analyzed to further validate the analytical model. Due to the stress singularity at the crack tip and the influence of far-field boundaries, the analysis is confined to the normalized distance interval *x*/*a*_0_ ∈ [2,7]. Figure 3b shows that within this range, the theoretical interfacial normal stress remains largely stable overall but displays a slight upward trend as *x*/*a*_0_ increases. In simulations, the variation in such curves is complicated. When *H*/*a*_0_ is 0.67, as the *x*/*a*_0_ increases, the interfacial normal stress undergoes an initial dip, a gradual recovery, and a subsequent decline. But once the value of *H*/*a*_0_ is greater than 0.67, the absolute value of the normal stress decreases as *x*/*a*_0_ increases. When the *H*/*a*_0_ ratio exceeds 1.5, the absolute value of the normal stress first decreases and then increases as *x*/*a_0_* increases. In addition, the larger the value of *H*/*a*_0_ is, the steeper these curves are. Compared with the FEA model, for the range of 0.8 to 1.0 (the neighbor of the crack tip), the analytical model can predict the change trend of interface normal stress with acceptable accuracy. To our surprise, the simulation results are always inconsistent with the theory under the condition that the value of *x*/*a*_0_ is large than 1.0 (i.e., the point close to infinity). Reexamining the boundary condition in the theoretical model, we identified that the stress boundaries at the infinity point on the interface are ignored (assuming to be zero). However, the huge difference in thermal expansion of bi-material leads to the inevitable stress at the terminal point on the interface. However, within the appropriate scope, this does not affect the overall trend.

## 4. Material Parameter Analyses

To evaluate interfacial failures caused by large CTE mismatches in solar cell modules, we examined the thermal properties of common encapsulants and solar cells. The analysis employs a bi-layer model with both the upper encapsulant film and the lower solar cell set to a uniform thickness of 150 μm. The model is subjected to a uniform heat flux of 0.05 W/m^2^ on its top surface, and the relevant material properties are provided in Table 1 (the parameters in Table 1 are referenced from [1,2,3,4]).

Figure 4a presents the distribution of interfacial normal stress as a function of the normalized distance *x*/*a*_0_. A stress concentration trend similar to that observed in the previous section is evident. The results indicate that the interfacial normal stresses remain largely unaffected by the type of solar cell, suggesting that minor variations in the thermal expansion coefficient of the stiff layer exert negligible influence compared to the overall material mismatch.

As illustrated in Figure 4b, while the stress concentration follows a consistent trend, the influence of different encapsulant materials on the interfacial normal stress varies significantly. Among the five commonly used encapsulant films paired with solar cells, the Thermoplastic Elastomer (TPSE)/solar cell combination exhibits the highest maximum interfacial normal stress, whereas the Polyolefins (PO)/solar cell pair shows the lowest, with values of −79.44 MPa and −35.37 MPa, respectively. Among these five material sets, the EVA/solar cell combination behaves very similarly to the Cover slip/solar cell pair, which is widely used in practical manufacturing. The remaining three groups display a distinct pattern. Given that the primary differentiating parameter among these encapsulants is their coefficient of thermal expansion (CTE), it can be inferred that the CTE is a key factor governing the severity of interfacial stress concentration.

Analysis of the five sets of curves reveals that the relationship between interfacial normal stress and the thermal expansion coefficient is more complex than a simple positive or negative correlation. For instance, among PO, EVA, and Cover slip, a smaller thermal expansion coefficient corresponds to a larger maximum interfacial normal stress. However, PDMS has a smaller thermal expansion coefficient than EVA, yet its maximum interfacial normal stress remains lower than that of PO. A similar inconsistency is observed for TPSE. These results suggest the presence of additional influencing factors. The materials considered in this study are constrained by practical engineering requirements, which preclude a strictly controlled-variable experimental design. Therefore, while the focus has been placed on thermal expansion properties, the role of elastic modulus may have been overlooked.

To better decouple the individual and combined effects of these two parameters, the following analysis focuses on the location along *x*/*a*_0_ where the interfacial normal stress reaches zero, referred to here as the “stress critical point”. For this purpose, we assume an ideal encapsulating material whose properties can be independently adjusted. Figure 5 illustrates the relationship between the maximum stress and the ratio of thermal expansion coefficients (or elastic moduli). The results indicate that when *E*_1_/*E*_2_ is held constant, the stress critical point increases linearly with *CTE*_1_/*CTE*_2_. Similarly, when *CTE*_1_/*CTE*_2_ is fixed, *E*_1_/*E*_2_ also shows a linear positive correlation with the critical stress point.

Based on practical engineering considerations, several realistic combinations of semiconductor and encapsulation layers were selected. Their elastic modulus and CTE ratios are presented in dimensionless form in Figure 6. Overall, the critical stress point exhibits a positive correlation with the elastic modulus ratio and a negative correlation with the CTE ratio. According to this trend, the critical stress point at *x*/*a*_0_ = 6.858 should be smaller than that at *x*/*a*_0_ = 6.843. However, the observed result contradicts this expectation. This discrepancy can be attributed to the fact that in real material combinations, the variation in *CTE*_1_/*CTE*_2_ typically remains within one order of magnitude, whereas the modulus ratio (*E*_1_/*E*_2_) can span several orders of magnitude. Consequently, the influence of the elastic modulus ratio tends to dominate in practical engineering scenarios.

Finally, the influence of layer thickness is examined. In actual production, the thickness of the cover slip or solar cell typically ranges from 100 μm to 300 μm. In the model shown in Figure 7, the solar cell thickness is fixed at 150 μm, while that of the cover slip varies between 100 μm and 250 μm. The results indicate that the interfacial normal stress initially decreases and then increases with the normalized distance *x*/*a*_0_. Moreover, the maximum interfacial normal stress shows a positive correlation with the thickness of the cover layer.

## 5. Conclusions

In this paper, we developed a simplified model for assessing thermal stress in solar cell laminates exhibiting significant property mismatch under thermal loads. By incorporating common encapsulation films and solar cell types, we investigated the coupled influence of thermal expansion and elastic properties. The model proves to be a robust and efficient tool for predicting stress concentration trends, thereby providing directly applicable guidelines for optimizing solar module design and manufacturing.

## 6. Prospects

This paper investigates interfacial phenomena in soft/hard composite materials under thermal loading, though the findings are established under idealized conditions. Key limitations include the following: (1) the model considers solely thermal loading and omits combined mechanical effects or pre-existing cracks; (2) material interfaces are assumed perfect, neglecting practical concerns such as interfacial contact resistance; (3) the analysis is confined to static crack configurations and does not extend to scenarios involving periodic crack propagation.

## Figures and Tables

**Figure 1 micromachines-16-01320-f001:**
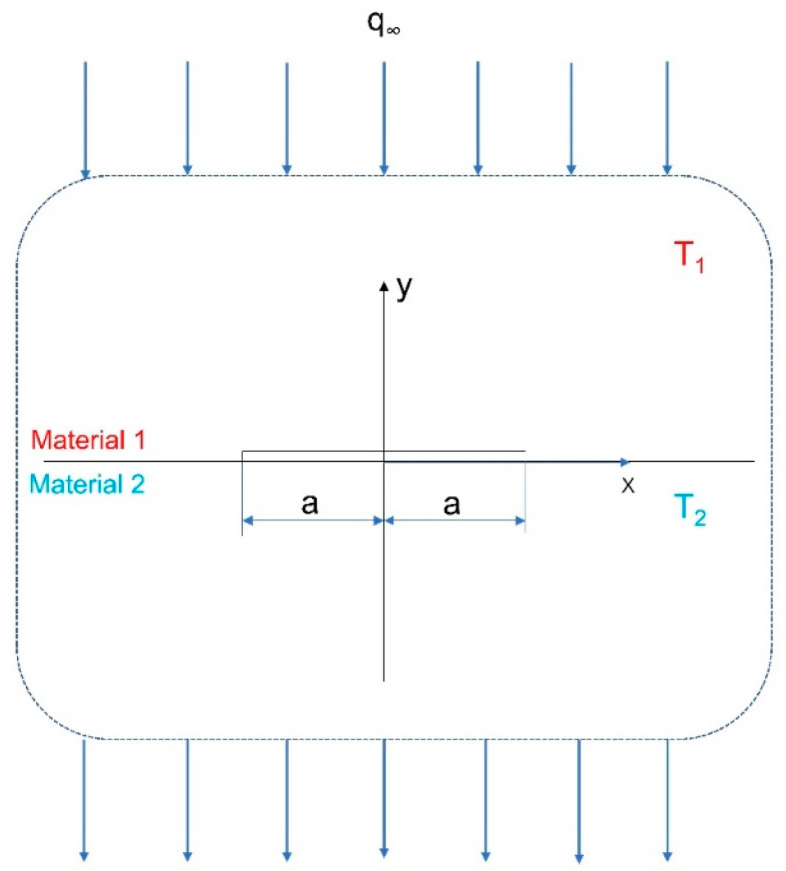
A Theoretical model with infinite boundaries.

**Figure 2 micromachines-16-01320-f002:**
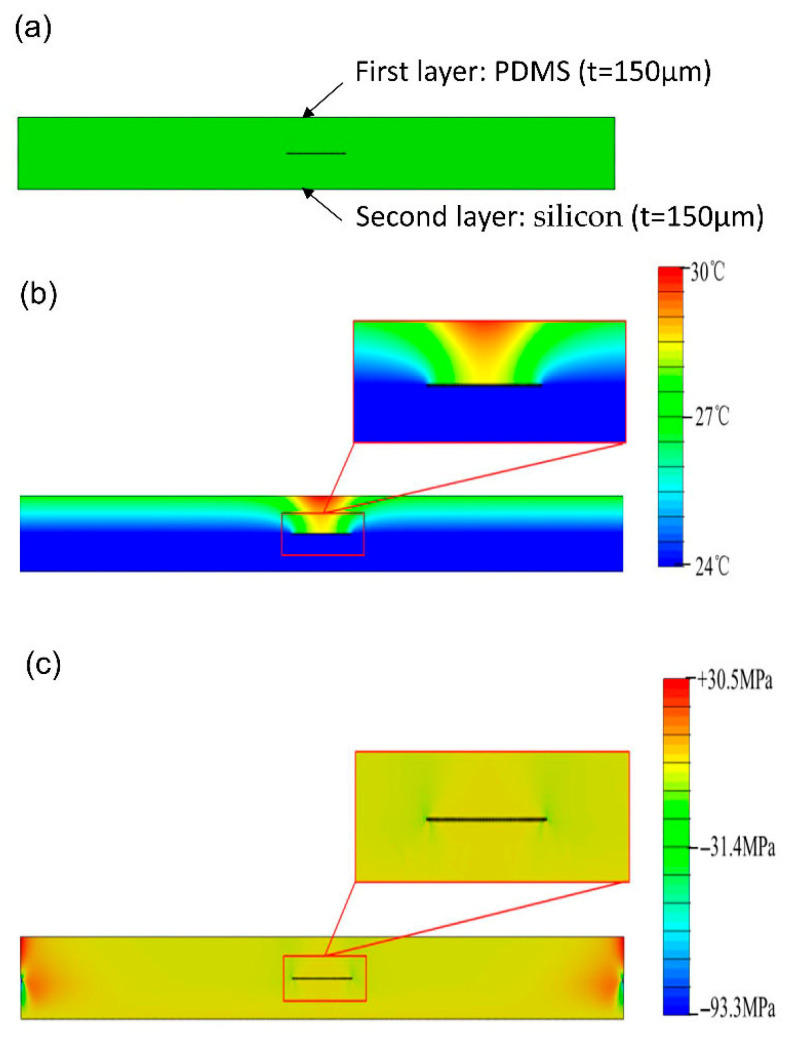
Simulation models for the bi-layer structure, (**a**) original state, (**b**) temperature distribution, and (**c**) normal stress distribution, respectively.

**Figure 3 micromachines-16-01320-f003:**
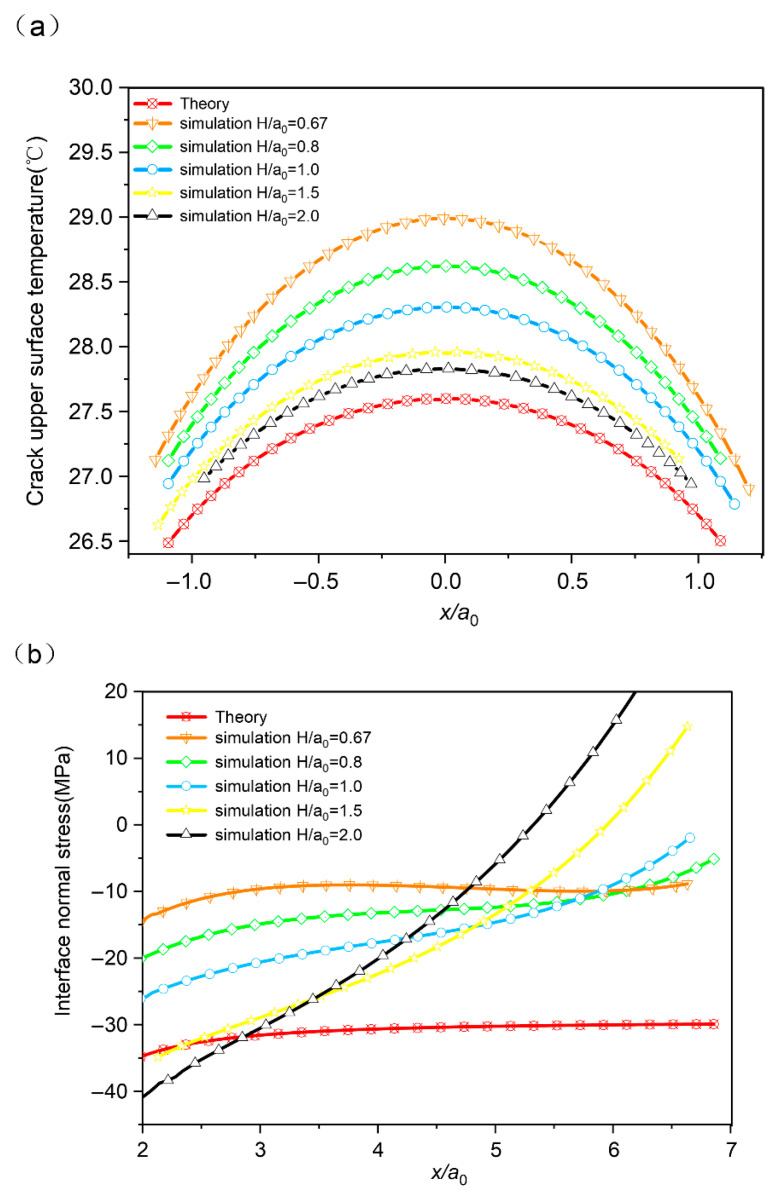
(**a**) Crack upper surface temperature and (**b**) interface normal stress for H/*a*_0_ ranging from 0.67 to 2.0, respectively.

**Figure 4 micromachines-16-01320-f004:**
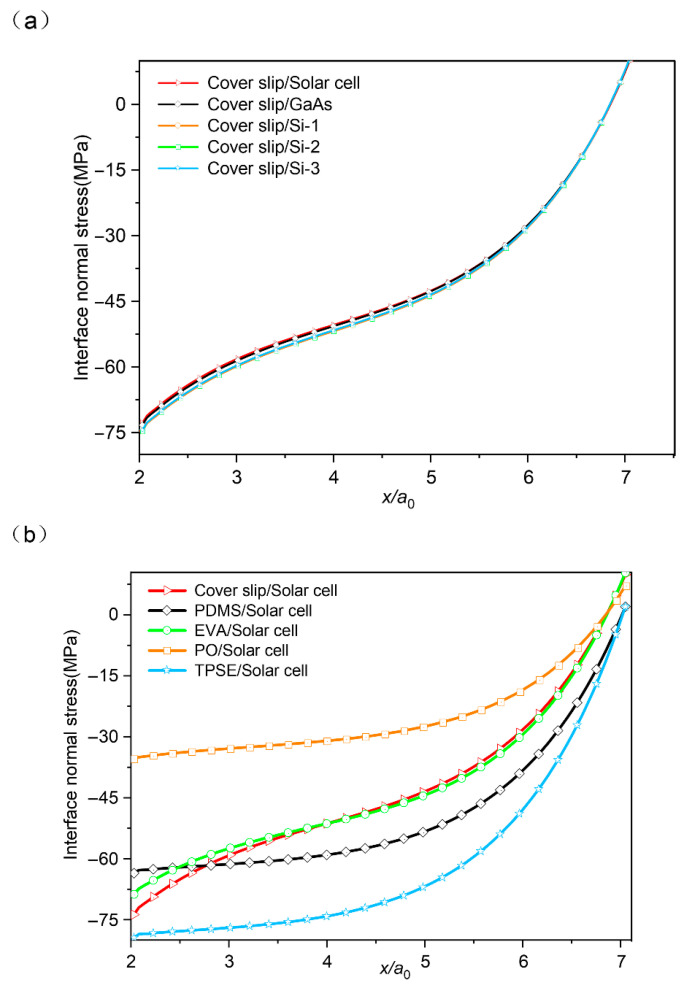
The interfacial normal stress profile: comparative analysis of the role of (**a**) the semiconductor layer and (**b**) the encapsulation layer.

**Figure 5 micromachines-16-01320-f005:**
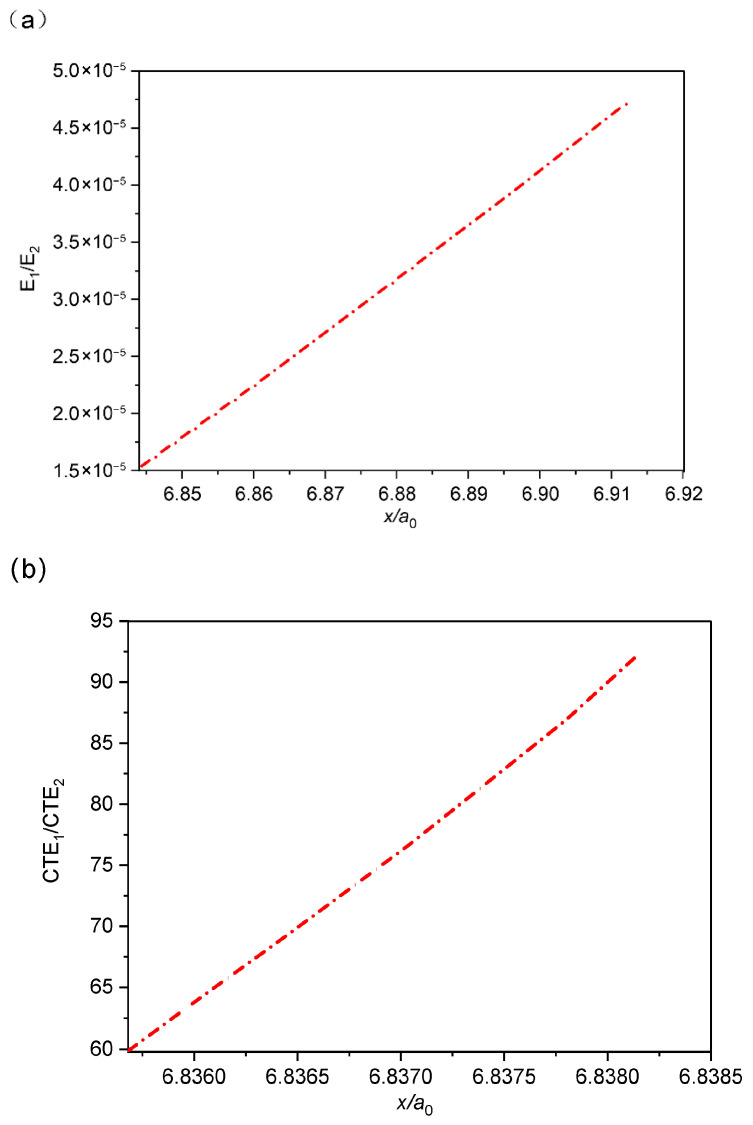
Critical stress point as a function of (**a**) elastic modulus ratio and (**b**) thermal expansion coefficient ratio.

**Figure 6 micromachines-16-01320-f006:**
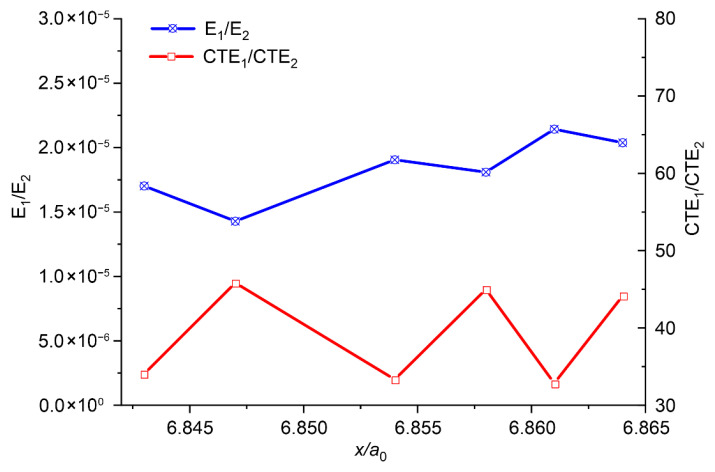
The influences of elastic modulus ratio and CTE ratio for engineering applications.

**Figure 7 micromachines-16-01320-f007:**
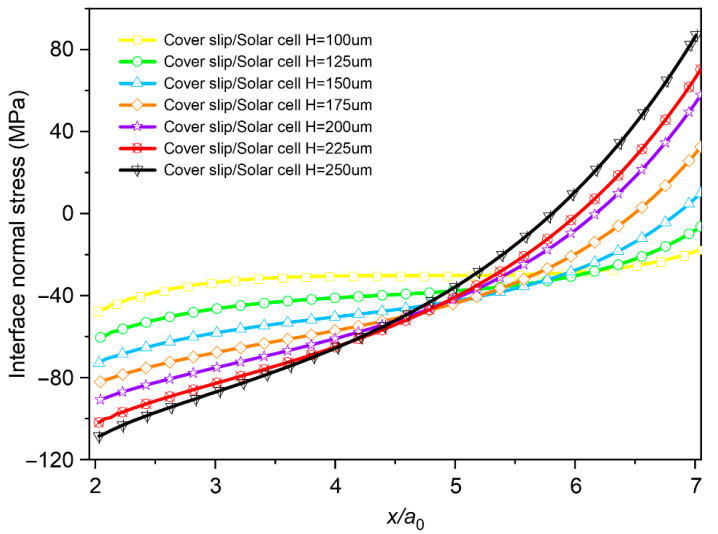
The influences of cover layer thickness on the interface normal stress.

**Table 1 micromachines-16-01320-t001:** List of CTE values for materials used in analyses.

Material	CTE/°C				
cover slip	2 × 10^−4^				
Solar cell	6 × 10^−6^				
GaAs		5.87 × 10^−6^			
Si				2.60 × 10^−6^	2.60 × 10^−6^
Si-1			2.61 × 10^−6^		
Si-2			2.92 × 10^−6^		
Si-3			3.34 × 10^−6^		
PDMS	2.15 × 10^−4^				
EVA			2.70 × 10^−4^		
PO					2.75 × 10^−4^
TPSE					2.65 × 10^−4^

## Data Availability

Data will be made available on request.

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
