# Peer review of "A Theoretical Solution for Analyzing Bi-Layer Structures with Differing Thermal Properties†"

_micromachines, 2025, doi:10.3390/mi16121320_

Round 1

Reviewer 1 Report

Comments and Suggestions for Authors

Review Comments for "A Theoretical Solution for Analyzing Bi-Layer Structures with Differing Thermal Properties"

Overall Evaluation

The manuscript presents a theoretical framework for analyzing bi-layer structures with distinct thermal properties, which addresses a relevant topic in thermal management and composite materials. The proposed solution demonstrates potential for advancing the understanding of heat transfer in layered systems, and the theoretical derivations appear rigorous. However, several critical issues related to figure/table clarity, notation consistency, and experimental design documentation need to be addressed to ensure reproducibility and readability. Below are detailed comments and recommendations for major revision.

Specific Comments and Recommendations

1. Figure 2: Missing Material and Structural Annotations

In Figure 2, which illustrates the simulation setup for the bi-layer structure, the authors have not provided essential details about the material types, structural dimensions (e.g., layer thicknesses, lateral sizes), or geometric parameters. This omission hinders readers’ ability to critically assess the simulation design or replicate the study.
Recommendation:

  • Annotate Figure 2 with explicit labels for: (i) the materials of each layer (e.g., "Layer 1: Aluminum; Layer 2: Polyimide"), (ii) key dimensions (e.g., "Thickness of Layer 1: ; Layer 2: "), and (iii) geometric features (e.g., "Lateral size: ").

  • Include a scale bar if applicable to clarify spatial dimensions.

2. Figure Readability: Font, Color, and Key Information Highlighting

Multiple figures (e.g., Figures 1, 3–5) suffer from poor readability due to small font sizes, low-contrast colors, and insufficient emphasis on key data. For instance, curve labels in Figure 3 are barely legible.
Recommendation:

  • Adhere to scientific figure standards: Use sans-serif fonts (e.g., Arial, Helvetica) with a minimum size of 8 pt (preferably 10–12 pt for labels).

  • Replace low-contrast colors with distinct, colorblind-friendly palettes (e.g., dark red, blue, green) and ensure line thicknesses ≥ 2 pt.

  • Highlight critical data (e.g., peak values, transition points) with bold symbols, annotations, or insets to guide readers’ interpretation.

3. Table 1: Ambiguous Notation [1]

In Table 1, which summarizes material properties or simulation parameters, the notation "[1]" is used without explanation. This ambiguity leaves readers uncertain whether [1] refers to a reference citation, a specific experimental condition, or a data source.
Recommendation:

  • Clarify the meaning of [1] in the table caption or footnotes (e.g., "[1] denotes data from Reference [X]; [2] indicates measured values").

  • Ensure all non-standard notations in tables are explicitly defined to avoid misinterpretation.

4. Unclear Definition of  in Figures

The normalized coordinate  appears in multiple figures (e.g., Figures 4–6) but is not defined in the figure captions or main text. Additionally, the material basis for the data (e.g., which specific bi-layer system the  values correspond to) is unspecified.
Recommendation:

  • Define  directly in the figure captions (e.g., ": lateral position; : characteristic length (e.g., layer thickness )").

  • Specify the material system for each dataset (e.g., "Data for  correspond to a bi-layer of SiOâ‚‚/PDMS with ").

Conclusion

The manuscript addresses a valuable problem but requires significant revisions to meet scientific reporting standards. The primary issues—unclear figure/table annotations, poor readability, and ambiguous notations—undermine the clarity and reproducibility of the work. With thorough revisions to address these points (as outlined above), the paper has strong potential for publication.

Decision: Major Revision.

Author Response

Please refer to the following file.

Reviewer 2 Report

Comments and Suggestions for Authors

In this work, the authors present a simplified practical approach for predicting thermal stress concentration trends in laminated solar cell structures that provides a useful insight for the design and fabrication of solar modules. The authors clearly explain the motivation and novelty in the Introduction. This theoretical approach is well presented and helps position the work within existing research.

However, despite the strong novelty several aspects of this work need to be addressed.

  1. Figure 3b shows that the interface normal stress increases with increasing x/a0 value for H/a0 values greater than 0.67. But in the discussion author described the opposite.

“But once the value of H/a0 is greater than 0.67, the normal stress decreases with the increase in x/a0.”

  1. Why didn’t the simulated interfacial stress follow the theoretical trend for a higher H/a0 value?
  2. The results shown in Figure 5 were obtained under the assumption of an ideal encapsulating material. How do the authors correlate these results for real encapsulating materials?
  3. What are the limitations of this approach? Authors should mention this in the manuscript.

Author Response

Please refer to the following file.

Round 2

Reviewer 1 Report

Comments and Suggestions for Authors

I have no futher questions.